# Assessing the Anti-Inflammatory and Antioxidant Activity of Mangiferin in Murine Model for Myocarditis: Perspectives and Challenges

**DOI:** 10.3390/ijms25189970

**Published:** 2024-09-16

**Authors:** Alexandra Popa, Lia-Oxana Usatiuc, Iuliu Calin Scurtu, Raluca Murariu, Alexandra Cofaru, Romelia Pop, Flaviu Alexandru Tabaran, Luciana Madalina Gherman, Dan Valean, Alexandru Cristian Bolundut, Rares Ilie Orzan, Ximena Maria Muresan, Andreea Georgiana Morohoschi, Sanda Andrei, Cecilia Lazea, Lucia Agoston-Coldea

**Affiliations:** 1Department of Internal Medicine, Iuliu Hatieganu University of Medicine and Pharmacy, 400347 Cluj-Napoca, Romania; 2Department of Pediatrics, Iuliu Hatieganu University of Medicine and Pharmacy, 400347 Cluj-Napoca, Romania; 3Department of Pathophysiology, Iuliu Hatieganu University of Medicine and Pharmacy, 400347 Cluj-Napoca, Romania; 4Department of Small Animal Internal Medicine, Faculty of Veterinary Medicine, University of Agricultural Sciences and Veterinary Medicine, 400372 Cluj-Napoca, Romania; 5Department of Anatomic Pathology, Faculty of Veterinary Medicine, University of Agricultural Sciences and Veterinary Medicine, 400372 Cluj-Napoca, Romania; 6Experimental Center, Iuliu Hatieganu University of Medicine and Pharmacy, 400012 Cluj-Napoca, Romania; 7Regional Institute of Gastroenterology and Hepatology “O. Fodor”, 400162 Cluj-Napoca, Romania; 8Department of Translational Medicine, Institute of Medical Research and Life Sciences—MEDFUTURE, Iuliu Hatieganu University of Medicine and Pharmacy, 400349 Cluj-Napoca, Romania; 9Department of Biochemistry, Faculty of Veterinary Medicine, University of Agricultural Sciences and Veterinary Medicine, 400372 Cluj-Napoca, Romania

**Keywords:** autoimmune myocarditis, experimental model, rats, myocardial inflammation, IL-1β, IL-6, TNF-α, nitro-oxidative stress markers, total antioxidant capacity, flavonoids, Mangiferin, corticosteroids, Trolox, echocardiography, left ventricular ejection fraction, hystopathology, inflammatory infiltrates

## Abstract

Myocarditis is a major cause of heart failure and death, particularly in young individuals. Current treatments are mainly symptomatic, but emerging therapies focus on targeting inflammation and fibrosis pathways. Natural bioactive compounds like flavonoids and phenolic acids show promising anti-inflammatory and antioxidant properties. Corticosteroids are frequently employed in the treatment of autoimmune myocarditis and appear to lower mortality rates compared to conventional therapies for heart failure. This study aims to explore the effects of Mangiferin on pro-inflammatory cytokine levels, nitro-oxidative stress markers, histopathological alterations, and cardiac function in experimental myosin-induced autoimmune myocarditis. The effects were compared to Prednisone, used as a reference anti-inflammatory compound, and Trolox, used as a reference antioxidant. The study involved 30 male Wistar–Bratislava rats, which were randomly divided into five groups: a negative control group (C−), a positive control group with induced myocarditis using a porcine myosin solution (C+), three groups with induced myocarditis receiving Mangiferin (M), Prednisone (P), or Trolox (T) as treatment. Cardiac function was evaluated using echocardiography. Biochemical measurements of nitro-oxidative stress and inflammatory markers were conducted. Finally, histopathological changes were assessed. At echocardiography, the evaluation of the untreated myocarditis group showed a trend toward decreased left ventricular ejection fraction (LVEF) but was not statistically significant, while all treated groups showed some improvement in LVEF and left ventricular fraction shortening (LVFS). Significant changes were seen in the Mangiferin group, with lower end-diastolic left ventricular posterior wall (LVPWd) by day 21 compared to the Trolox group (*p* < 0.001). In the first week of the experiment, levels of interleukins (IL)-1β, IL-6, and tumour necrosis factor (TNF)-α were significantly higher in the myosin group compared to the negative control group (*p* < 0.001, *p* < 0.001, *p* < 0.01), indicating the progression of inflammation in this group. Treatment with Mangiferin, Prednisone, and Trolox caused a significant reduction in IL-1β compared to the positive control group (*p* < 0.001). Notably, Mangiferin resulted in a superior reduction in IL-1β compared to Prednisone (*p* < 0.05) and Trolox (*p* < 0.05). Furthermore, Mangiferin treatment led to a statistically significant increase in total oxidative capacity (TAC) (*p* < 0.001) and a significant reduction in nitric oxide (NOx) levels (*p* < 0.001) compared to the negative control group. Furthermore, when compared to the Prednisone-treated group, Mangiferin significantly reduced NOx levels (*p* < 0.001) and increased TAC levels (*p* < 0.001). Mangiferin treatment significantly lowered creatine kinase (CK) and aspartate aminotransferase (AST) levels on day 7 (*p* < 0.001 and *p* < 0.01, respectively) and reduced CK levels on day 21 (*p* < 0.01) compared to the untreated group. In the nontreated group, the histological findings at the end of the experiment were consistent with myocarditis. In the group treated with Mangiferin, only one case exhibited mild inflammatory infiltrates, represented by mononucleated leukocytes admixed with few neutrophils, with the severity graded as mild. Statistically significant correlations between the grades (0 vs. 1–2) and the study groups have been highlighted (*p* < 0.005). This study demonstrated Mangiferin’s cardioprotective effects in autoimmune myocarditis, showing reduced oxidative stress and inflammation. Mangiferin appears promising as a treatment for acute myocarditis, but further research is needed to compare its efficacy with other treatments like Trolox and Prednisone.

## 1. Introduction

Myocarditis presents a diagnostic challenge due to its heterogeneous clinical manifestations and increasing prevalence, remaining a leading cause of morbidity and mortality, affecting approximately 1% to 7% of individuals worldwide [1]. 

In the pathophysiological process, myocarditis involves both innate and adaptive immune responses, with each contributing to different phases of the disease. The acute phase is driven by the innate immune system, particularly through cytokine and chemokine release, while the subacute and chronic phases engage the adaptive immune system, including T cells. If inflammation is not fully resolved, the disease may progress to a chronic condition, leading to dilated cardiomyopathy (DCM) and heart failure. Various immune cells, such as macrophages and T-helper cells, play key roles in this progression, with their effects modulated by cytokines and other signaling molecules. Despite advances, the complete immunological mechanisms of myocarditis are not fully understood, warranting further research to develop targeted therapies [2]. 

Regarding autoimmune myocarditis, it could be one of the forms with a poor prognosis, which can progress to dilated DCM in advanced stages. Notably, up to 40% of heart failure cases in individuals under the age of 40 are caused by autoimmune myocarditis [3]. 

Current therapeutic approaches for myocarditis are primarily symptomatic, focusing on managing heart failure and rhythm disturbances. Nonetheless, emerging strategies aim to develop therapies that directly target the inflammatory and fibrotic pathways implicated in myocarditis pathogenesis. The use of medicinal herbs for treating various health conditions, including cardiovascular diseases, is gaining increasing recognition [3,4,5]. 

Flavonoids, a class of polyphenolic compounds, are known for their antihypertensive, anti-atherosclerotic, platelet inhibitory, and endothelial protective effects. Recent studies have highlighted their roles in ischemia/reperfusion myocardial injuries, dependent on their structural characteristics, through possible interactions with specific pharmacological targets [3,6,7,8]. In the physiopathology of acute myocarditis, inflammation plays a key role characterized by the activation of nuclear factor κB (NF-κB) signaling pathways and the release of pro-inflammatory cytokines, including tumor necrosis factor-alpha (TNF-α) and interleukins (IL): IL-1, IL-6, IL-12, and IL-23 [2,9]. Oxidative stress has a crucial role in the progression of the autoimmune process, with the suppression of antioxidant defenses and prolonged nitro-oxidative stress contributing to cardiac remodeling and inflammatory cardiomyopathy. Through their hydroxyl groups, flavonoids can neutralize free radicals, directly scavenge superoxide anions, and deactivate oxygen-derived reactive species such as peroxynitrite [10,11,12,13]. 

Mangiferin (1,3,6,7-tetrahydroxyxanthone-C2-β-D-glucoside) is a glucosyl xanthone and bioactive compound isolated from *Mangifera indica* species that has demonstrated a wide range of therapeutic effects, including antioxidant, anti-inflammatory, cardioprotective, antibacterial, antiviral, and immunomodulatory effects [14]. The mechanisms underlying Mangiferin’s effects include inhibition of NF-κB activation, upregulation of transforming growth factor (TGF)-β, and downregulation of cyclooxygenase (COX)2 transcriptional activity, which prevent the expression of inflammatory mediators such as inducible nitric oxide synthase and TNF-α [9,11,12,15]. 

Trolox (6-hydroxy-2,5,7,8-tetramethylchroman-2-carboxylic acid), a cell-permeable, water-soluble vitamin E derivative, exhibits potent antioxidant and anti-inflammatory properties [16]. Studies have demonstrated its role in the NF-κB signaling pathway and its ability to prevent lipid peroxidation (LPO), oxidative stress, and apoptosis by reducing reactive oxygen species (ROS) production [17,18]. 

Corticosteroids, such as Prednisone, are widely used for their immunosuppressive effects in the treatment of myocarditis, particularly in cases with an autoimmune etiology. The interaction between corticoids and the glucocorticoid (GC) receptor leads to the activation of PI3-kinase and the protein kinase Akt in the heart, modulating nitric oxide release and ultimately exerting a cardioprotective effect [19]. Some studies showed that the administration of corticosteroids has not been associated with a significant reduction in mortality compared to conventional heart failure treatment, although it may improve left ventricular (LV) function [20,21]. However, current studies have shown that standard immunosuppressive therapy is recommended to be administered long term and tailored to the patient [22,23]. Caforio and collaborators, in their very recent study results, support the effectiveness and safety of such therapy in immune-mediated myocarditis [23]. 

One of the multiple pathways of the fibrosis process is depicted schematically in Figure 1.

This study aims to investigate the effects of Mangiferin, Prednisone, and Trolox on cardiac functional parameters, pro-inflammatory cytokine profiles, nitro-oxidative stress markers, and histopathological changes in experimental autoimmune myocarditis (EAM). The goal is to identify potential adjuvant therapies for the treatment of this condition.

## 2. Results

### 2.1. Echocardiographic Results

Cardiac structure and function were assessed by measuring the following parameters: interventricular septum systolic thickness (IVSs), left ventricular posterior wall diastolic thickness (LVPWd), left ventricular diameter in diastole (LVDd) and in systole (LVDs), left ventricular ejection fraction (LVEF), and left ventricular fractional shortening (LVFS) (Figure 2).

Significant changes were observed in the Mangiferin group. On day 2, the LVDs was significantly smaller compared to the positive control group (*p* < 0.05). By day 21, differences in LVPWd were reported in the Mangiferin group, with significantly lower values compared to the Trolox group (*p* < 0.001). 

In the Prednisone group, significant differences in IVSs were observed on day 2 compared to both the positive control group and the Trolox group (*p* < 0.05). Additionally, LVDd and LVDs were significantly lower compared to both control groups (*p* < 0.05).

In the Trolox group, increased LVPWd values were reported on day 2 compared to the positive control group. LVDd and LVDs were significantly lower compared to the negative control group (*p* < 0.05) and the positive control group (*p* < 0.01), respectively. By day 21, LVDs values remained lower compared to the negative control group, and there was a significant increase in LVPWd thickness from day 2 to day 21 (*p* < 0.01). Additionally, LVFS was significantly improved compared to the negative control group (*p* < 0.05). 

In the untreated myocarditis group, there was a trend towards a decrease in LVEF. In all three treated groups, there was a tendency for improvement in both LVEF and LVFS, although these changes did not reach statistical significance (Figure 3). 

### 2.2. Biochemical Results 

On the first day of the experiment, no significant differences were observed in nitric oxide (NOx) and total antioxidant capacity (TAC) levels between groups, except in the Trolox-treated group, which showed a statistically significant increase in both parameters compared to the untreated group (*p* < 0.001 for NOx, *p* < 0.01 for TAC). 

By day 7, TAC levels were significantly decreased (*p* < 0.01), and NOx levels were significantly increased (*p* < 0.001) in the Trolox group compared to the negative control group. Mangiferin treatment led to a statistically significant increase in TAC (*p* < 0.001) and a significant reduction in NOx levels (*p* < 0.001) compared to the negative control group. Furthermore, when compared to the Prednisone-treated group, Mangiferin significantly reduced NOx levels (*p* < 0.001) and increased TAC levels (*p* < 0.001). No significant differences were observed between the Trolox and Mangiferin-treated groups. 

On the final day of the experiment, significant differences between the negative control group and the myosin-induced myocarditis group persisted for both TAC (*p* < 0.01) and NOx (*p* < 0.001). Statistically significant differences among the Mangiferin-treated group and the positive control, Prednisone, and Trolox groups remained evident at the end of the experiment for both parameters (Table 1).

We observed no significant differences in non-specific cardiac injury markers between groups on the first day of the experiment, except for aspartate aminotransferase (AST) levels, which were lower in the Mangiferin-treated group compared to the Trolox group (*p* < 0.05). Creatine kinase (CK) and AST levels were significantly higher in the myosin group compared to the negative control group on day 7 (*p* < 0.01) and on the last day of the experiment (*p* < 0.01).

Mangiferin treatment significantly lowered CK and AST levels on day 7 (*p* < 0.001 and *p* < 0.01, respectively) and reduced CK levels on day 21 (*p* < 0.01) compared to the untreated group. Prednisone also led to a significant reduction in CK levels (*p* < 0.001), while Trolox significantly lowered both CK (*p* < 0.01) and AST (*p* < 0.05) levels on day 7 of the experiment.

Statistically significant differences in AST levels were also observed on days 7 and 21 in the Mangiferin-treated group compared to the Prednisone group (*p* < 0.05), with Mangiferin showing a more pronounced lowering effect (Table 2).

Starting from the first day of the experiment, myosin administration led to a significant increase in IL-1β levels (*p* < 0.001). Treatment with Mangiferin, Prednisone, and Trolox caused a significant reduction in IL-1β compared to the positive control group (*p* < 0.001). Notably, Mangiferin resulted in a superior reduction of IL-1β compared to Prednisone (*p* < 0.05) and Trolox (*p* < 0.05). TNF-α levels were higher in the Prednisone-treated group compared with the Mangiferin-treated group (*p* < 0.05). By day 7, levels of IL-1β, IL-6, and TNF-α were significantly higher in the myosin group compared to the negative control group (*p* < 0.001, *p* < 0.001, *p* < 0.01), indicating the progression of inflammation in this group. Mangiferin significantly reduced IL-1β (*p* < 0.01), IL-6 (*p* < 0.05), and TNF-α (*p* < 0.01) levels compared to the untreated group. Prednisone and Trolox caused a mild reduction in IL-1β (*p* < 0.05), while Trolox slightly reduced IL-6 levels (*p* < 0.05), and Prednisone mildly decreased TNF-α levels (*p* < 0.05). 

On the final day of the experiment, IL-1β and IL-6 levels were significantly lower in the Mangiferin-treated group compared to the myosin group (*p* < 0.001). Moreover, when compared to Trolox, Mangiferin demonstrated a superior reduction in IL-1β and IL-6 levels (*p* < 0.01, *p* < 0.05) (Table 3). 

In both the positive control group and the Mangiferin-treated group, Pearson’s correlation analysis revealed no significant correlations between oxidative stress markers (NOx and TAC), cardiac injury markers (CK and AST), and inflammatory cytokines (IL-1β, IL-6, and TNF-α).

### 2.3. Histopathological Results

Histological sections were analyzed on day 21 to assess myocardial inflammation and injury severity, focusing on inflammatory infiltrates and fibrosis. Inflammatory severity was categorized as minimal, mild, moderate, or severe based on the presence of lymphocytes, macrophages, and, occasionally, neutrophils, along with fibrous tissue formation.

In the C+ group, histological findings on day 21 were indicative of myocarditis. The lesions primarily showed minimal leukocyte infiltrates (n = 4), composed of mononuclear cells mixed with a few neutrophils. In three of these cases, minimal fibrosis (histological score 1) was associated with the inflammatory infiltrate. No significant findings were observed in one individual from this group.

In the group treated with Mangiferin, only one case exhibited mild inflammatory infiltrates, including mononuclear leukocytes and neutrophils, with the severity graded as mild.

No histopathological changes were observed in the groups treated with Prednisone and Trolox, indicating the potential effectiveness of these treatments in preventing histological signs of myocarditis. We summarize the histological findings in Figure 4.

Statistically significant correlation between the severity assessment (0 vs. 1–2) and the study groups has been highlighted (*p* < 0.005).

The histopathological images obtained for comparative analysis are illustrated in Figure 5.

Mortality in the studied groups was quantified as follows: no deaths were recorded in the C− group or the M group. In the untreated positive control group, the mortality rate was 33.3% (n = 2). In the P group, the recorded mortality was 50% (n = 3), while in the T group, mortality occurred in 16.6% of cases (n = 1). Notably, applying the χ^2^ test for independence revealed a statistically significant *p*-value only for the Trolox-treated group (*p* = 0.03).

## 3. Discussion

Autoimmune myocarditis remains a controversial subject due to the limited diagnostic and therapeutic options available. The gold standard for diagnosis is endomyocardial biopsy, but this procedure is not always reliable or feasible for patients with suspected acute myocarditis. The pathophysiology of EAM is complex, involving the excessive production of ROS and oxidative stress. This process triggers the release of inflammatory cytokines and chemokines, which attract leukocytes to the heart tissue, leading to cardiomyocyte damage through necrosis or apoptosis. This damage contributes to ventricular dilation and impaired systolic function [24,25].

To better understand this pathology and improve diagnostic accuracy, experimental models like the murine model of myocarditis induced by porcine myosin have been developed. By understanding these processes, specific therapeutic options can be generated that may later be applied to human patients [26].

In our study, we used a standardized model of myocarditis in adult Wistar–Bratislava rats, chosen for their genetic background, which allows for the extrapolation of results to humans, as well as their ease of handling and the ability to collect sufficient biological samples. This study contributes to the understanding of the cardioprotective effect of Mangiferin in EAM induced in rats by improving myocardial function and reducing oxidative stress and inflammation. Recent studies have highlighted the role of flavonoids in mediating cardioprotection in various cardiac conditions, including autoimmune myocarditis, doxorubicin-induced cardiotoxicity, myocardial infarction, and ischemia-reperfusion injury [6,27,28].

We have demonstrated that the administration of Mangiferin, Trolox, and Prednisone can reduce LV dimensions and wall thickness in rats with autoimmune-induced myocarditis, with a more pronounced effect observed in the Mangiferin group. Left ventricular wall thickening in the setting of acute myocarditis is usually transient and has been reported in a few studies over the last decades. These studies demonstrated that the thickness of the posterior wall is greatest in days 1–3 after myocarditis onset and improves to near normal during the convalescent phase [29,30,31]. Other studies supported the hypothesis that this process is caused by interstitial edema [31]. As we have demonstrated, Trolox has a lower effect than Mangiferin in reducing the inflammatory process, and the persistence of increased thickness of the posterior LV wall on day 21 could be attributed to persistent inflammation.

These changes could likely be attributable to the reduction in inflammatory cell infiltration, resulting in the normalization of ventricular contraction. This is evidenced by increased LVEF and LVFS, along with decreased LVD in the treated groups, particularly evident on day 21 of evaluation. Such findings may indicate a positive outcome in preventing left ventricular remodeling and the progression to DCM and heart failure. Similar findings were reported by Jiang et al. in heart failure-induced rats [10]. Song et al. also demonstrated restoration of cardiac function in rats with induced myocardial fibrosis by transverse aortic constrictions by preventing the rise in the left ventricular end-systolic and diastolic end-volumes and decreasing of the LV mass, showing that Mangiferin can attenuate heart dysfunction and protect against myocardial remodeling via inflammation inhibition [8]. Restoration of hemodynamic functions and preservation of ventricular contraction was more evident in the Trolox group, although trends toward increased LVEF and LVFS were detected in all treated groups. We considered the obtained values, although they did not have statistical significance, with the main limitation being the sample size. Additionally, the LVEF is an observer-dependent measurement and difficult to quantify in rats due to their small heart size and increased heart rate, so obtaining an optimal image and calculating diameters or volumes with the axes as equal as possible are considered challenging aspects. Echocardiography was merely a suggestive tool for assessing induced acute myocarditis, not a diagnostic one in our study, and is not as accurate for quantifying kinetic impairment resulting from inflammation, edema, or fibrosis, aspects that could be more accurately described by cardiac magnetic resonance imaging.

For the study protocol, we aimed to select a dose of Mangiferin that would achieve both its antioxidant and anti-inflammatory effects. According to the literature, the minimum dose reported to produce these effects is 10 mg/kg [32]. On the other hand, the milder effects of Mangiferin on systolic function parameters in our study may be attributed to the lower dose used (25 mg/kg) compared to higher doses (40 mg/kg) that have significantly improved these parameters in rats with myocardial ischemia-reperfusion injury [10]. This comparison does not take into account the cumulative effects of Mangiferin, which was administered for 21 days in our study, versus the 15-day administration of higher doses in the previously mentioned study. These findings are supported by Zheng et al., whose results indicated significant improvements in LV dimensions and contractility in myocardial infarction model rats treated with Mangiferin in a dose-dependent manner [12].

Studies on the cardioprotective effects of Trolox in experimental myocarditis are limited. However, we observed a decrease in LV dimensions and an increase in LVFS compared to the negative control group, suggesting that Trolox may have a beneficial effect in reducing the damaging effects of myocardial inflammation [33].

In comparison to both control groups, LV dimensions also diminished in the Prednisone group. However, these results should be interpreted cautiously due to the small sample size. Randomized controlled trials of corticosteroid administration for myocarditis have shown that treatment may improve LVEF in the first three months, but the statistical significance diminishes by the end of observation due to substantial heterogeneity [21,34].

In our experiment, we measured serum levels of the pro-inflammatory markers TNF-α, IL-1β, and IL-6. Mangiferin administration significantly reduced these markers, while Trolox had milder effects on IL-6. These results are consistent with previous studies [16,18]. The cardioprotective mechanism of Prednisone has been well-documented, particularly through its interaction with glucocorticoid receptors and modulation of antioxidant effects [17]. Our study showed that Prednisone administration reduced serum levels of TNF-α and IL-1β. The observed variations in IL-1, IL-6, and TNF-α levels across our experimental groups reflect the degree of inflammation and immune response, potentially indicating either the intensity or resolution of these processes. While our statistically significant results suggest that these differences are unlikely to be attributable to chance, we were unable to directly establish their clinical or biochemical significance. It is essential to evaluate whether these biomarker fluctuations have substantive implications for patient outcomes or disease progression. To confirm their utility as prognostic indicators or to assess treatment efficacy, long-term studies are required to validate their potential in clinical settings.

Regarding oxidative stress, our findings on the action of Mangiferin align with those in the literature [9]. Moreover, our study demonstrated the superiority of Mangiferin over Prednisone, with significantly higher TAC values and lower NOx levels in the Mangiferin-treated group compared to the Prednisone-treated group.

Histopathological confirmation of myocarditis in our study was based on the presence of inflammatory infiltration and fibrosis in heart tissues, consistent with findings from other studies using the experimental acute myocarditis rat model [26,33]. At the end of the experiment, the treated groups showed significantly improved myocardial architecture, with inflammation observed in only one case in the Mangiferin-treated group. The severity of lesions was significantly higher in the positive control group that did not receive any treatment. These beneficial effects are likely linked to the strong anti-inflammatory and antioxidant properties of Mangiferin and Trolox, as demonstrated in other cardiovascular disease models [10,16,18]. Although no inflammatory infiltrate was found in the histopathological evaluation of the prednisone-treated group, 50% of the rats (n = 3) died during the last week of the experiment. The cause of death remains unclear, although myocarditis was excluded as the determining factor based on histopathological results. It is important to note that this could be a limitation of glucocorticoid administration in experimental models, potentially due to their other effects rather than their immunosuppressive action [14].

However, since both the reference antioxidant (Trolox) and the reference anti-inflammatory agent (Prednisone) have significantly improved histopathological findings, and Mangiferin, which has both antioxidant and anti-inflammatory properties, also showed a notable improvement in histopathological appearance, this highlights its promising potential as an adjuvant treatment for myocarditis.

There are some limitations to our study. The relatively small number of animals in each group could limit the statistical power of the findings. This could especially impact the reliability of conclusions drawn from subgroups, such as those treated with Prednisone, where significant mortality occurred without a clear explanation. This highlights the importance of carefully considering potential side effects when evaluating treatment outcomes. In our study, mortality could introduce bias and influence the interpretation of results, particularly in assessing the efficacy and safety of Prednisone. Another limitation is the lower dose of Mangiferin compared to other studies that used higher doses. This may have limited the ability to observe the full potential of Mangiferin’s cardioprotective effects, making it difficult to compare directly with other research. The study was limited to a 21-day observation period, which may not be sufficient to fully understand the long-term effects of the treatments, particularly in terms of cardiac remodeling, chronic inflammation, and potential side effects. While the study measured several important biomarkers (e.g., TNF-α, IL-1β, IL-6, TAC, NO), other relevant biomarkers or additional mechanistic studies might have provided more insight into the pathways through which Mangiferin, Trolox, and Prednisone exert their effects. Although the rat model of myocarditis is widely used and provides valuable insights, it may not fully replicate the human condition. Differences in immune system function, heart physiology, and response to treatments between rodents and humans limit the direct applicability of the findings to clinical practice. The study focused primarily on biochemical and histopathological outcomes. Including additional functional assessments, such as cardiac output, exercise tolerance, or heart rate variability, could have provided a more comprehensive evaluation of the treatment effects.

## 4. Materials and Methods 

The current study was approved by the local ethics committee of “Iuliu Hatieganu” University of Medicine and Pharmacy, Cluj Napoca (decision number 100/30.05.2024).

### 4.1. Animals

The research involved 30 Wistar–Bratislava albino male rats (*Rattus norvegicus*), aged 16 weeks and weighing 200 ± 50 g at the beginning of the study. The rats were obtained from the Animal Department of the Faculty of Medicine, Iuliu Hatieganu University of Medicine and Pharmacy Cluj-Napoca. The animals underwent a two-week acclimatization period and were housed in labeled polypropylene cages by group under controlled conditions: temperature (24 ± 2 °C), humidity (60 ± 5%), and a 12 h light/dark cycle. The animals had unrestricted access to standard food and water (ad libitum). Body weights were monitored throughout the experiment to adjust substance dosages accordingly.

### 4.2. Study Design

The rats were randomly divided into five groups: negative control group (C−): healthy nontreated rats; positive control group (C+): untreated rats with induced myocarditis; Mangiferin-treated group (M): rats with myocarditis treated with Mangiferin; Prednison-treated group (P): rats with myocarditis treated with Prednisone; Trolox treated group (T): rats with myocarditis treated with Trolox (Figure 6). For the C− group, 1 mL of 0.9% saline solution was injected subcutaneously. The C+, M, P, and T groups received a porcine myosin,calcium activated from porcine heart-buffered aqueous glycerol solution 0.1–0.5 units/mg (Sigma-Aldrich Chemical Co., St. Louis, MO, USA, code M0531) to induce acute myocarditis. This solution was prepared by dissolving one vial of porcine myosin (0.84 mL) in 1.16 mL of Freund’s adjuvant, and 0.05 mL (0.25 mg myosin/100 g body weight) was injected subcutaneously on days 0 and 7.

The Mangiferin (Bertin Bioreagent, Montigny le Bretonneux, France, code 22360) solution was prepared by dissolving 100 mg of Mangiferin powder(≥98% purity) in 20 mL of dimethyl sulfoxide (DMSO). The rats in the M group received 0.5 mL/100 g body weight (2.5 mg Mangiferin/100 g body weight) via gavage from days 2 to 21.

The Prednisone solution was prepared by dissolving one 5 mg Prednisone tablet in 10 mL of saline. The rats in the P group received 0.5 mL/100 g body weight (0.25 mg Prednisone/100 g body weight) via gavage from days 2 to 21. The Trolox solution was administered at a dose of 20 mg/100 g body weight via gavage from days 2 to 21 in the T group.

### 4.3. Echocardiography

Echocardiographic evaluations were performed using an Ultrasonix Medical corporation SonixTablet (Richmond, BC, Canada) system equipped with a S12-4MHz linear probe transducer. The animals were shaved bilaterally at the thorax, and a generous amount of echocardiographic gel was applied. Rats were positioned in dorsal recumbency, as described by Brown et al. [35]. The examination began with the parasternal long-axis 4-chamber view to subjectively assess contractility and the presence of pericardial fluid. Next, the short-axis view at the level of the papillary muscles was used to obtain M-mode measurements of the left ventricular walls and chamber dimensions, specifically the interventricular septum (IVS), left ventricular internal diameter (LVID), and left ventricular posterior wall (LVPW) during systole and diastole, across three cardiac cycles.

Echocardiography was performed on days 2 and 21 by two operators, with both present at each evaluation. Anesthesia was induced with intraperitoneal administration of Xylazine (5 mg/kg) and Ketamine (75 mg/kg).

### 4.4. Biochemical Measurements

Under light anesthesia with 10% ketamine and 2% xylazine (in a 2:1 ratio), blood samples (1 mL) were collected from the retro-orbital plexus 24 h after myocarditis induction and on days 7 and 21, using heparinized tubes. Plasma was obtained by centrifugation at 4 °C for 20 min at 1620× *g*, then transferred to Eppendorf tubes and stored at −80 °C. At the end of the experiment, the animals were sacrificed using an overdose of anesthetics.

Nitro-oxidative stress was assessed by measuring nitric oxide synthesis (NOx) and total oxidative capacity (TAC). Plasma levels of NOx, TAC, inflammatory cytokines (TNF-α, Il-1B, Il-6) and non specific cardiac injury markers (CK, AST) were determined using the enzyme-linked immunosorbent assay (ELISA) technique by employing specific assay kits (Elabscience Biotechnology Inc., Houston, TX, USA). Spectroscopic measurements were performed using a microplate reader (Spectrostar Nano, BMG Labtech, Ortenberg, Germany).

### 4.5. Histopathological Examination

Collected tissues were fixed in 10% formalin for 48 h. After fixation, tissues were dehydrated through a series of increasing ethanol concentrations, cleared in xylene, and infiltrated with paraffin (melting point: 58 °C) for 5 h. Sections, 2 μm thick, were cut from the paraffin blocks using a rotary microtome. Before staining, sections were deparaffinized by immersion in xylene (three times, 2 min each) and hydrated through decreasing ethanol concentrations (three immersions in 100% ethanol for 2 min each, followed by 2 min in 95% ethanol, and a brief dip in 70% ethanol). To complete hydration, the slides were rinsed under running tap water at room temperature for at least 2 min. Staining began with a 5 min immersion in hematoxylin solution to highlight the cell nuclei, followed by a 5 min rinse under running tap water to remove excess stain. The slides were then stained with eosin solution for 2 min to stain the cytoplasm and enhance the specimen’s contrast. The sections were then dehydrated through multiple immersions in ethanol, starting with approximately 20 dips in 95% ethanol, a 2 min incubation in the same solution, and finishing with two 2 min immersions in 100% ethanol. To remove any residual substances, the sections were washed in xylene three times for 2 min each. Slides were then mounted with Permount, coverslipped, and prepared for microscopic examination. Fibrosis quantification was performed using Masson’s trichrome stain (MT), following a previously described protocol [36]. The histological grade of myocardial fibrosis was evaluated using a method initially described by Rezkalla et al. [37] and later adapted by Li et al. [38]. The scoring system used was as follows: a score of “0” indicated no significant fibrosis; a score of “1” was assigned when fibrosis occupied less than 25% of the total cardiac section area; a score of “2” indicated fibrosis covering 25% to 50% of the area; and a score of “3” was given when fibrosis covered between 50% and 75% of the total histological cardiac section area. Histological analysis was performed using an Olympus BX51 microscope, with bright-field images captured by an Olympus SP350 digital camera and processed with Olympus cellSens software (Version 3.1). Microscopic changes were classified according to the International Harmonization of Nomenclature and Diagnostic (INHAND) standards [39], using a grading system where 0 = no significant change, 1 = minimal, 2 = mild, 3 = moderate, and 4 = severe.

### 4.6. Statistical Methods

Data were collected using Microsoft Excel 2021 and analyzed with IBM spss v26.0. Normality of data distribution was tested using Kolmogorov–Smirnov and Shapiro–Wilk tests. Differences of mean values between two groups was verified using the T-test and ANOVA for three or more groups. Correlation between quantitative variables was verified using Pearson’s correlation test. We assessed the mortality rate among different treatment groups using χ^2^ test for independence. The threshold for statistical significance was considered *p* < 0.05.

Cardiac morphometric parameters were expressed as mean ± standard deviation. Group comparisons were conducted using One-Way ANOVA with Tukey’s and Bonferroni’s multiple comparisons tests, with *p* < 0.05 as the threshold for statistical significance. Statistical analysis was performed using GraphPad Prism Software (version 8.0.1; GraphPad Software Inc., La Jolla, CA, USA).

## 5. Conclusions

The results of our study highlight the cardioprotective effects of Mangiferin in experimental autoimmune myocarditis by reducing oxidative stress and inflammation, and by improving the histopathological changes and the overall myocardial function. The results were comparable to the results obtained after Prednisone and Trolox administration. However, further research is essential to fully understand its efficacy and validate the results.

## Figures and Tables

**Figure 1 ijms-25-09970-f001:**
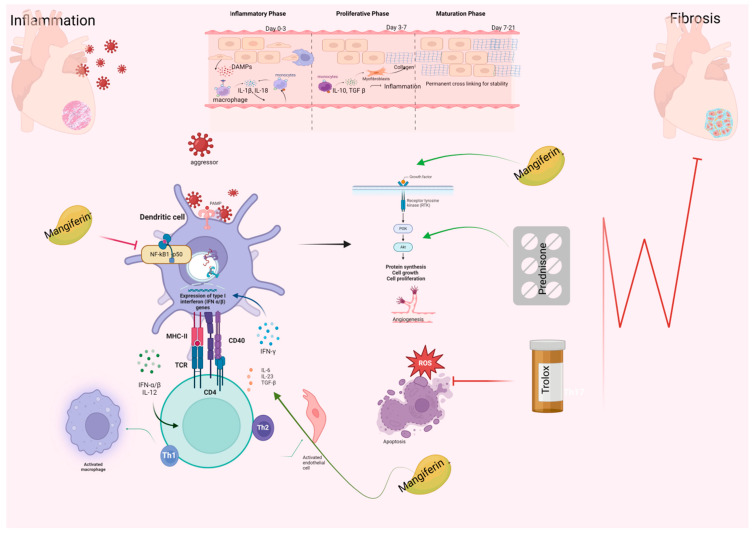
A proposed schematic representation of the progression from inflammation to fibrosis in heart tissue following exposure to an external pathogen. Dendritic cells recognize the pathogen, triggering the activation of the nuclear factor (NF)-κB/p50 signaling pathway, leading to the expression of type I interferon (IFN-α/β) genes. This interaction further activates T helper (Th) cells, with Th1 promoting inflammation via cytokines such as IFN-γ and Th2, mediating anti-inflammatory responses through interleukin (IL)-6, IL-23, transforming growth factor (TGF)-β, and Th17, contributing to chronic inflammation and fibrosis. Macrophages also amplify the inflammatory response through cytokine release. Mangiferin inhibits the NF-κB pathway, thereby reducing immune cell activation. Trolox reduces oxidative stress by decreasing reactive oxygen species (ROS) and preventing apoptosis, while Prednisone is used to decrease inflammation. Additionally, the activation of receptor tyrosine kinases (RTKs) initiates the PI3K/Akt pathway, driving cell growth, proliferation, and angiogenesis, ultimately leading to tissue remodeling and fibrosis.

**Figure 2 ijms-25-09970-f002:**
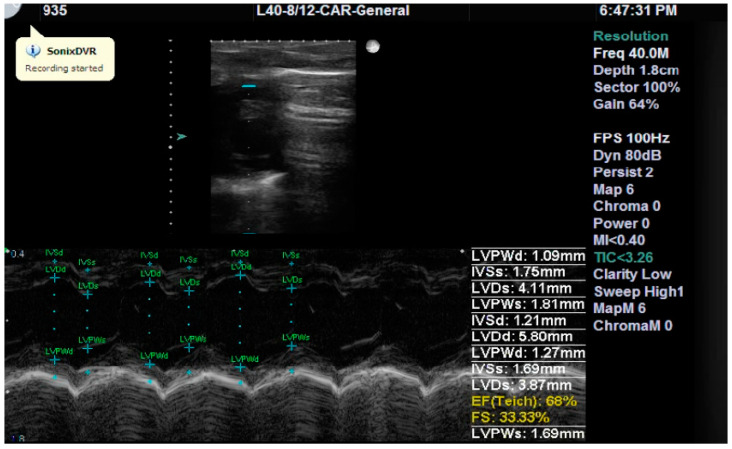
A representative echocardiographic image (parasternal short axis view at the level of the papillary muscles) showing the measurements taken on day two from a subject in the Mangiferin group for illustrative purposes.

**Figure 3 ijms-25-09970-f003:**
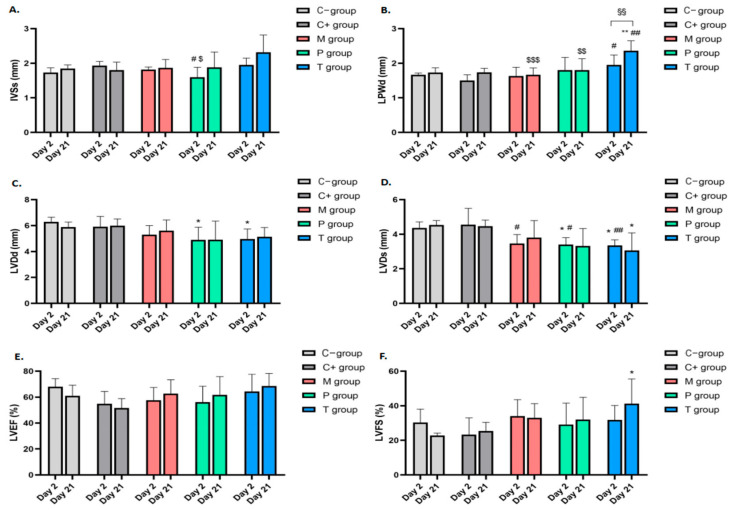
Echocardiographic evaluation of left ventricular function in EAM rats. (**A**). Interventricular septum systolic thickness (IVSs); (**B**). left ventricular posterior wall diastolic thickness (LVPWd); (**C**). left ventricular diastolic diameter (LVDd); (**D**). left ventricular systolic diameter (LVDs); (**E**). left ventricular ejection fraction (LVEF); (**F**). left ventricular fractional shortening (LVFS). * Significance vs. C− group for the corresponding day, *p* < 0.05; ** significance vs. C− group for the corresponding day, *p* < 0.01; # significance vs. C+ group for the corresponding day, *p* < 0.05; ## significance vs. C+ group for the corresponding day, *p* < 0.01; $ significance vs. T group for the corresponding day, *p* < 0.05; $$ significance vs. T group for the corresponding day, *p* < 0.01; $$$ significance vs. T group for the corresponding day, *p* < 0.001; §§ day 2 vs. day 21 for the corresponding group, *p* < 0.01.

**Figure 4 ijms-25-09970-f004:**
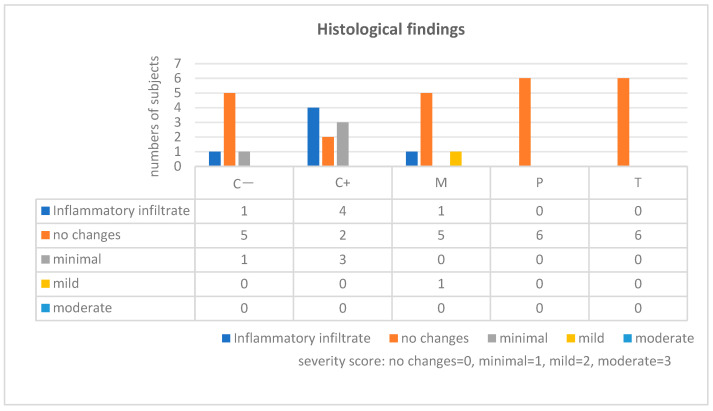
Incidence of myocarditis-suggestive lesions compared across the studied groups and the severity score assessement. C−: negative control group, C+: positive control group with myocarditis induced, M: myocarditis group treated with Mangiferin, P: myocarditis group treated with Prednisone, T: myocarditis group treated with Trolox.

**Figure 5 ijms-25-09970-f005:**
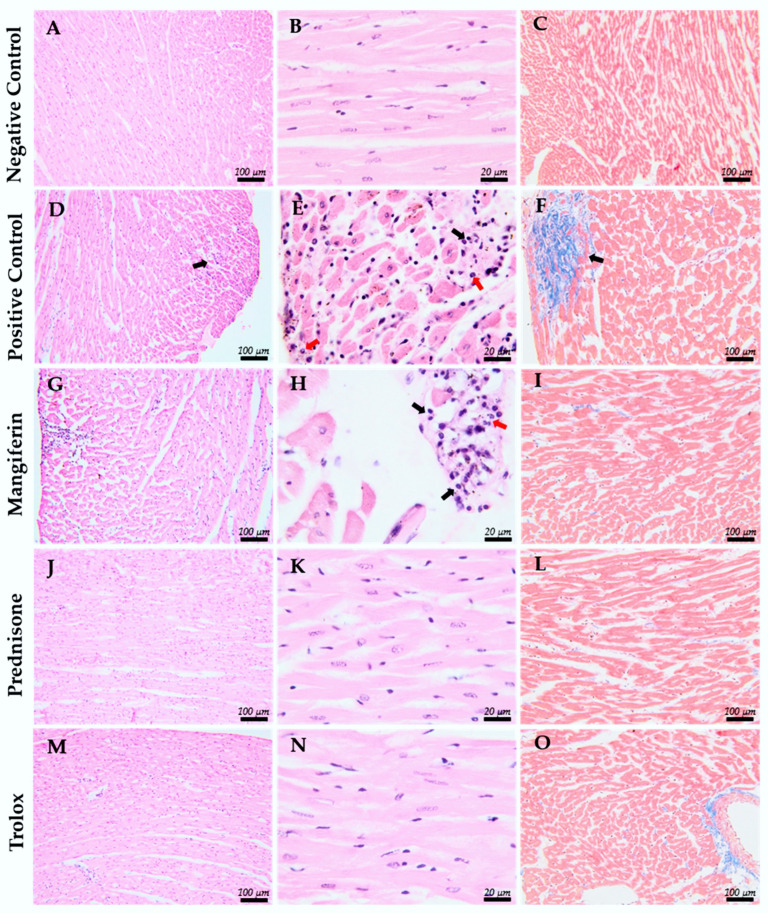
Histopathological images of myocardial tissue. (**A**–**C**) Negative control group—no significant findings are observed. (**D**–**F**) Positive control group with induced myocarditis (without treatment)—showing multifocal inflammatory infiltrate consisting of mononuclear leukocytes (black arrows), a few neutrophils (red arrows), and focal fibrosis ((**F**), indicated by the arrow). (**G**–**I**) Mangiferin-treated group—in one individual, there is focal inflammatory infiltrate consisting of mononuclear leukocytes (black arrows) with a few neutrophils (red arrows). Significant fibrosis is absent. (**J**–**L**) Prednisone-treated group—no significant findings observed. (**M**–**O**) Trolox-treated group—no significant findings observed. H&E stain, Obx20: images (**A**,**D**,**G**,**J**,**M**); Obx100: images (**B**,**E**,**H**,**K**,**N**); TM stain, Obx20: images (**C**,**F**,**I**,**L**,**O**).

**Figure 6 ijms-25-09970-f006:**
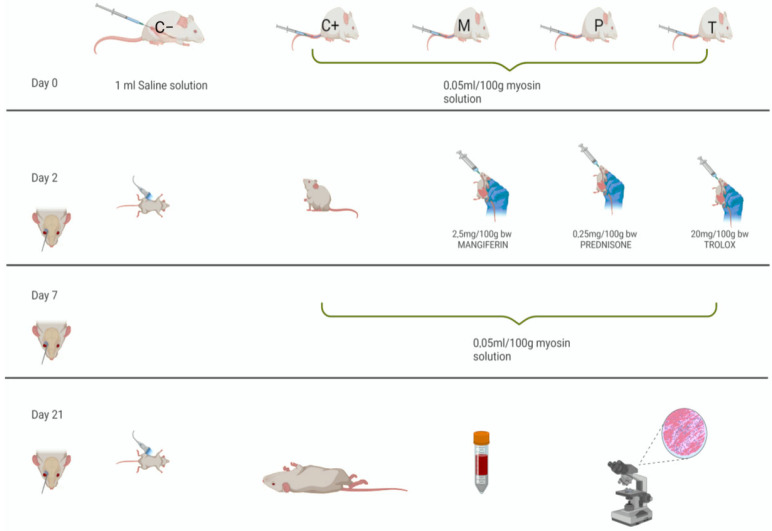
Distribution of the experimental groups, treatments, and investigations. The negative control group (C−) received a subcutaneous injection of 1 mL of 0.9% saline solution. Myocarditis was induced through porcine myosine administration on days 0 and 7 in the C+, M, P, and T groups by subcutaneous administration of 0.25 mg/100 g body weight of myosin solution (0.05 mL). The M group was treated with Mangiferin (2.5 mg/100 g body weight), the P group received Prednisone (0.25 mg/100 g body weight), and the T group was administered Trolox (20 mg/100 g body weight). The treatments were given via gavage from day 2 to day 21. Echocardiography was performed on days 2 and 21, and blood samples were collected on days 2, 7, and 21.

**Table 1 ijms-25-09970-t001:** Effects of Mangiferin on oxidative stress markers in rat myosin-induced myocarditis.

Parameters		Control	MYO	MYO + Mangiferin	MYO + Prednisone	MYO + Trolox
TAC (mM/L)	Day 1	4.4578 ± 0.47840	3.8298 ± 0.77534	5.3990 ± 2.51656	0.54806 ± 0.24510	7.6120 1.08881 ^###^
	Day 7	4.4578 ± 0.47840	3.5163 ± 0.36650 **	23.7990 ± 1.54047 ^###,*fff*^	4.6398 ± 0.44866 ^##^	24.9750 0.86038 ^###^
	Day 21	4.5150 ± 0.47737	3.6316 ± 0.41136 **	17.9914 ± 5.92671 ^###,*f*^	4.7478 ± 0.49852 ^##^	24.4950 2.07838 ^###^
NOx (μM/L)	Day 1	23.9091 ± 1.46233	24.7780 ± 1.32288	24.6468 ± 1.93246 *^a^*	23.6460 ± 1.50291	22.4434 1.08271 ^##^
	Day 7	23.9091 ± 1.46233	78.7268 ± 9.23031 ***	24.5228 ± 1.56627 ^###,*fff*^	60.0000 ± 12.02775 ^#^	25.2330 3.31222 ^###^
	Day 21	23.7728 ± 1.50088	120.0474 ± 60.89025 ***	22.7570 ± 1.26222 ^##^	34.2778 ± 13.69287 ^#^	22.0875 0.79721 ^##^

Values are expressed as mean ± SD (standard deviation). ** *p* < 0.01, *** *p* < 0.001; ^#^ vs. MYO: ^#^ *p* < 0.05, ^##^
*p* < 0.01, ^###^ *p* < 0.001; *^f^* vs. MYO + Prednisone: *^f^ p* < 0.05, *^fff^ p* < 0.001; *^a^* vs. MYO + Trolox: *^a^ p* < 0.05, MYO—myosin; TAC—total antioxidant capacity; NOx—nitrites and nitrates.

**Table 2 ijms-25-09970-t002:** Effects of Mangiferin on non-specific cardiac injury markers in rat myosin-induced myocarditis.

Parameters		Control	MYO	MYO + Mangiferin	MYO + Prednisone	MYO + Trolox
CK (U/L)	Day 1	45.2129 ± 4.01864	50.1628 ± 6.03302	54.6068 ± 3.55391	48.4108 ± 2.81584	48.4002 ± 4.72820
	Day 7	45.2129 ± 4.01864	72.5850 ± 4.70365 **	45.0002 ± 8.57386 ^###^	49.4698 ± 5.25098 ^###^	54.8575 ± 6.69585 ^##^
	Day 21	44.9895 ± 4.23595	71.1572 ± 15.75079 **	54.6068 ± 3.55391 ^#^	59.0058 ± 10.29713	52.1910 ± 3.71126 ^#^
AST (U/L)	Day 1	115.1777 ± 11.17335	130.5426 ± 22.42771	123.3760 ± 9.94124 *^a^*	128.4422 ± 7.60099	145.8846 ± 15.67173
	Day 7	124.2777 ± 12.16335	151.4463 ± 16.93884 ***	118.2890 ± 11.61918 ^##,*f*^	137.4253 ± 9.91660	123.4505 ± 8.64327 ^#^
	Day 21	114.4499 ± 11.71454	137.8176 ± 19.61112 ***	119.5748 ± 13.54963 *^f^*	140.3000 ± 4.02823	128.4973 ± 6.50378

Values are expressed as mean ± SD (standard deviation). ** *p* < 0.01, *** *p* < 0.001; ^#^ vs. MYO: ^#^ *p* < 0.05, ^##^ *p* < 0.01, ^###^ *p* < 0.001; *^f^* vs. MYO + Prednisone: *^f^ p* < 0.05; *^a^* vs. MYO + Trolox: *^a^ p* < 0.05 MYO—myosin; CK, creatine kinase, AST, aspartate aminotransferase.

**Table 3 ijms-25-09970-t003:** Effects of Mangiferin on inflammatory markers in rat myosin-induced myocarditis.

Parameters		Control	MYO	MYO + Mangiferin	MYO + Prednisone	MYO + Trolox
IL-1B pg/mL	Day 1	25.7374 ± 3.59190	74.8364 ± 5.97322 ***	22.3636 ± 1.03652 ^###,*f,a*^	24.9818 ± 1.40601 ^###^	29.6364 ± 5.19018 ^###^
	Day 7	25.7374 ± 3.59190	120.8182 ± 62.62460 ***	26.3091 ± 2.52884 ^##,*aa*^	26.3182 ± 2.01578 ^#^	49.5909 ± 1.63384 ^#^
	Day 21	24.7273 ± 2.06162	103.0909 ± 38.85446 ***	24.9091 ± 5.58185 ^##,*aa*^	26.4091 ± 1.50938 ^#^	49.3409 ± 15.46388 ^#^
IL-6 pg/mL	Day 1	34.3333 ± 7.5154	33.3232 ± 1.94075	37.6222 ± 4.78010	47.6000 ± 26.19533	35.9778 ± 5.41055
	Day 7	34.3333 ± 7.51542	63.1389 ± 11.85001 ***	47.9111 ± 7.31496 ^#^	51.8333 ± 6.17808	55.8333 ± 15.43085
	Day 21	36.4167 ± 4.46177	119.8889 ± 59.69573 **	38.1111 ± 5.53329 ^##,*a*^	53.4167 ± 25.25400	40.6111 ± 4.69085 ^#^
TNF-α pg/mL	Day 1	62.5238 ± 6.26783	57.4000 ± 1.82500	59.0286 ± 6.56086	63.8286 ± 2.83275 ^##^	59.3429 ± 4.34906
	Day 7	62.5238 ± 6.26783	94.1071 ± 23.31254 **	62.4000 ± 3.88141 ^##^	65.1429 ± 9.22139 ^#^	69.6071 ± 14.36567
	Day 21	63.5357 ± 5.86224	68.4286 ± 9.44371	61.8571 ± 7.85065	58.5000 ± 3.74983	62.8571 ± 5.01155

Values are expressed as mean ± SD (standard deviation). ** *p* < 0.01, *** *p* < 0.001; ^#^ vs. MYO: ^#^
*p* < 0.05, ^##^ *p* < 0.01, ^###^ *p* < 0.001; *^f^* vs. MYO + Prednisone: *^f^*
*p* < 0.05; ^*a*^ vs. MYO + Trolox: *^a^*
*p* < 0.05, *^aa^ p* < 0.01; MYO + myosin; IL-, interleukine; TNF-α, tumour necosis factor-α.

## Data Availability

The authors confirm that the data supporting the findings of this study are available within the article.

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
