# Peer review of "Assessing the Anti-Inflammatory and Antioxidant Activity of Mangiferin in Murine Model for Myocarditis: Perspectives and Challenges"

_ijms, 2024, doi:10.3390/ijms25189970_

Round 1

Reviewer 1 Report

Comments and Suggestions for Authors

Manuscript # ijms-3179864

Title: EFFECTS OF MANGIFERIN IN EXPERIMENTALLY INDUCED MYOCARDITIS IN RATS

In this study, Alexandra Popa et al investigated the effects of Mangiferin, Prednisone, and Trolox on cardiac functional parameters, pro-inflammatory cytokine profiles, nitro-oxidative stress markers, and histopathological changes in experimental autoimmune myocarditis. The authors investigate the potential adjuvant therapies for the treatment of this condition. The manuscript is well written and is original since particularly focus on its anti-inflammatory and antioxidant properties. The main findings include: all treatment significantly improved cardiac dysfunction in myocarditis-induced rats as well as inflammatory markers, in addition those treatments reduced the myocardial inflammatory infiltration and fibrosis.

Major comments

1)      The title is not completely aligned with the study because it only refers to Mangiferin, while the study also evaluated the effect of Prednisone and Trolox. I suggest that title should be modify according to the study

2)      It is important to show the mortality rate for all groups

3)      In methods the histological evaluation is described as semiquantitative which is acceptable. However, the results of histological findings are showed as number of animals (individuals) presenting or not the inflammatory infiltrate. I suggest quantifying the inflammatory infiltrate by using immunohistochemistry which will even allow to determine different inflammatory cells.

4)      The quality of figure 4 should be improve. Please also consider query # 3

5)      Representative pictures of histological findings look to different magnifications. On the other hand, the magnification of the representative pictures should be state, and a bar magnification should be included in the pictures. Please modify.

6)      Authors highlight the infiltration of neutrophils but since the end point of the protocol is 4 weeks (chronic evolution), should be expected that more lymphocytes and/or macrophages infiltrate the myocardium instead of neutrophils.   

7)      The arrows of the histological pictures show the inflammatory infiltration but do not mark the neutrophils as legend say. Additionally, fibrosis is not show in H-E representative pictures.

8)      In results the authors state “The lesions included minimal mononuclear infiltrates (n=3), one case with mononuclear cells mixed with a few neutrophils and fibrosis (n=1)….” But the fibrosis was not quantified although it was considered as determine of cardiac damage. I strongly suggest quantifying myocardial fibrosis.

9)      The manuscript conclusion could be strengthened. The statement that “The findings supports the utility of echocardiography as a valuable tool….” is irrelevant for the results presented in the manuscript and because it has been previously demonstrated the use of echocardiography for studying cardiovascular remodeling and function in rodents.

Author Response

Dear Reviewer,

The authors of the original article “EFFECTS OF MANGIFERIN IN EXPERIMENTALLY INDUCED MYOCARDITIS IN RATS” express deep gratitude to you for agreeing to review our work. Thank you very much for the critical, but professional analysis of our manuscript and for the remarks made, which we gratefully accept. The authors have made the necessary changes to the text in accordance with the comments made. The changes made have been marked in red in the new version of the manuscript.

Reviewer 1

Comment 1:      The title is not completely aligned with the study because it only refers to Mangiferin, while the study also evaluated the effect of Prednisone and Trolox. I suggest that title should be modify according to the study.

Response 1: We thank the reviewer for the observation. The title refers only to Mangiferin because Prednisone was used as a reference anti-inflammatory agent and Trolox was used as a reference antioxidant compounds, since their effect are already known. Mangiferin was the principal focus of our study. Therefore, we have changed the title: Assessing  the anti-inflammatory and antioxidant power of Mangiferin compared with Prednisone and Trolox in experimental autoimmune myocarditis: perspectives and challenges 

Also, we have reformulated the objectives: This study aims to explore the effects of Mangiferin on pro-inflammatory cytokine levels, nitro-oxidative stress markers, histopathological alterations and cardiac function in experimental myosin-induced autoimmune myocarditis. The effects were compared to Trolox, used as a reference antioxidant, and to Prednisone, used as a reference anti-inflammatory compound.

Comment 2:    It is important to show the mortality rate for all groups

Response 2: We thank you for the observation. The mortality rate in our study was not mentioned in the manuscript because during the experiment the manipulation of the animals for daily gavaje, blood withdrawal, echocardiography  performance (that needed sedation and analgesia) could have influenced the survival of the animals. Otherwise, we inserted a phrase with the mortality rate in each group, as you requested. Mortality in the studied groups was quantified as follows: no deaths were recorded in the C- group and the Mangiferin group. In the untreated positive control group, mortality was 33.3% (n=2). In the Prednisone-treated group, the recorded mortality was 50% (n=3), while in the Trolox-treated group, mortality occurred in 16.6% of cases (n=1). Notably, applying the χ2 test for independence  revealed a statistically significant p-value only for the Trolox-treated group (p=0.03).

Comment 3: In methods the histological evaluation is described as semiquantitative which is acceptable. However, the results of histological findings are showed as number of animals (individuals) presenting or not the inflammatory infiltrate. I suggest quantifying the inflammatory infiltrate by using immunohistochemistry which will even allow to determine different inflammatory cells.

Response 3: Thank you for your valuable feedback regarding the quantification of inflammatory cells in our study. We have carefully considered your recommendation and have revised Figure 4 accordingly. The figure has been improved to include not only the incidence of lesions per group but also the severity scores of the lesions. We believe this addition provides a more comprehensive representation of the data and enhances the overall clarity of our findings. We understand the suggestion to use immunohistochemistry (IHC) for a more precise identification of specific inflammatory cell types. However, we chose to use a semiquantitative method based solely on hematoxylin and eosin (H&E) staining for several reasons. Firstly the objective of our study was to provide a general assessment of inflammation, rather than to distinguish between specific cell subtypes, which aligns with the capabilities of H&E staining. While IHC can indeed provide more detailed information about specific cell types, it requires additional resources and was beyond the scope of our current study. The H&E-based approach we employed allowed us to achieve reliable semiquantitative data while maintaining consistency with similar studies in the field.

Comment 4:    The quality of figure 4 should be improve. Please also consider query # 3

Response 4: Thank you for the suggestion,  we have improved and changed the figures.

Comment 5:     Representative pictures of histological findings look to different magnifications. On the other hand, the magnification of the representative pictures should be state, and a bar magnification should be included in the pictures. Please modify.

Response 5: Thank you for your valuable feedback. We have carefully addressed your suggestions regarding the histological image panel. Specifically, we have ensured that similar magnification levels are used across all groups to facilitate direct comparison, and we have added scale bars to each image for clarity. Additionally, we have enhanced the panel by including Masson's trichrome images to better highlight the extent of fibrosis. Arrows have also been added to clearly identify the different leukocyte subtypes. We believe these changes significantly improve the quality and interpretability of the images.

Comment 6:  Authors highlight the infiltration of neutrophils but since the end point of the protocol is 4 weeks (chronic evolution), should be expected that more lymphocytes and/or macrophages infiltrate the myocardium instead of neutrophils.  

Response 6: Thank you for your insightful feedback. We have revised the manuscript in response to your input. As you correctly pointed out, at 21 days, the inflammatory cell infiltrate in myosin-induced myocarditis is indeed mixed, consisting primarily of mononuclear leukocytes with fewer neutrophils. We have re-evaluated this aspect and corrected the relevant information throughout the manuscript to ensure accuracy.We appreciate your valuable contribution to improving our work. 

Comment 7:  The arrows of the histological pictures show the inflammatory infiltration but do not mark the neutrophils as legend say. Additionally, fibrosis is not show in H-E representative pictures.

Response 7: Thank you for your feedback. As detailed in our response to comment number 5, we have revised the manuscript accordingly. The histological panel has been redone with the requested adjustments: arrows have been added to identify key features, and fibrosis is now clearly highlighted with the inclusion of Masson's trichrome images. These changes were made to enhance the clarity and comparability of the data presented.

Comment 8:     In results the authors state “The lesions included minimal mononuclear infiltrates (n=3), one case with mononuclear cells mixed with a few neutrophils and fibrosis (n=1)….” But the fibrosis was not quantified although it was considered as determine of cardiac damage. I strongly suggest quantifying myocardial fibrosis.

Response 8: Thank you for your constructive feedback. We have adjusted our manuscript as per your suggestions. Specifically, the quantification of fibrosis using Masson's trichrome stain has been added to the results. Additionally, we have updated the Materials and Methods section to include the relevant text and references to clarify the procedures used for fibrosis assessment.

Comment 9:   The manuscript conclusion could be strengthened. The statement that “The findings supports the utility of echocardiography as a valuable tool….” is irrelevant for the results presented in the manuscript and because it has been previously demonstrated the use of echocardiography for studying cardiovascular remodeling and function in rodents.

Response 9: We thank the reviewer for the observation. We have changed the conclusion with: The results of our study are highlighting the cardioprotective effects of Mangiferin in experimental autoimmune myocarditis, by reducing oxidative stress and inflammation, by improving the histopathological changes and the overall myocardial function. The results were comparable to the results obtained after Prednisone and Trolox administration. However, further research is essential to fully understand its efficacy and to validate the results.

We appreciate your input, which has helped us improve the clarity and completeness of our work.

Reviewer 2 Report

Comments and Suggestions for Authors

I have read with interest the manuscript entitled “Effects of Mangiferin in experimentally induced myocarditis in rats” by Popa et al.

This is an experimental study on the effects of various drugs on an experimental models of myocarditis; 30 mice were divided into 5 groups: a negative control group, a positive control group with induced myocarditis, and 3 groups with myocarditis receiving Mangiferin, Prednisone or Trolox. Measured outcomes were echocardiographic, biochemical and histological parameters.

The major results are that 1) IL-1β levels were lower after treatment with Mangiferin, Prednisone and Trolox (with Mangiferin resulting in the most relevant reduction), 2) Mangiferin also led to an increased total oxidative capacity and reduced nitric oxide levels.

The Authors’ conclusions are that Mangiferin has cardioprotective effects in autoimmune myocarditis, by reducing oxidative stress and inflammatory markers.

The study explores an interesting topic, and the manuscript is clearly written; the images are also informative and well structured. In addition, the results may have a clinical implication.

Nevertheless, the study raises the following concerns to be addressed.

MAJOR ISSUES

-   The fact that LVEF was not significantly lower in the myocarditis untreated group than the other groups (only a “trend” was observed, line 176) should be discussed more in depth. This is likely the main reason why echocardiographic features were not significantly improved by the different treatments (again, only a “trend” towards LVEF improvement was observed, line 177). To be sure that myocarditis was correctly induced in all groups, did some animal undergo myocardial biopsy before the end of the experiment (i.e. day 2)? In addition: “Line 280. In 1 case no inflammatory infiltrates were observed.”  This raises serious concerns about the fact that myocarditis was really induced in these models. Spontaneous healing of myocarditis is a likely reason, but it is not discussed here; actually, this may happen in >50% of human myocarditis, without any treatment. In my opinion, the Authors should focus on this aspect to reinforce the value of the overall findings.

-   There is a lack of consideration of recent literature clearly supporting the use of steroids in human virus-negative myocarditis; a large amount of literature has been published showing that steroid therapy has short and long-term benefits in terms of both overall survival and strong clinical outcomes such as LVEF improvement (see: 1) Chimenti Cet al. Immunosuppressive therapy in virus-negative inflammatory cardiomyopathy: 20-year follow-up of the TIMIC trial. Eur Heart J. 2022, 2) Caforio ALP et al. Long-term efficacy and safety of tailored immunosuppressive therapy in immune-mediated biopsy-proven myocarditis: A propensity-weighted study. Eur J Heart Fail. 2024); steroid therapy has also been included among the latest ESC recommendations in heart failure in humans. Please amend erroneous sentences both in the abstract (lines 36-37), introduction (line 126) and discussion sections.

-   The statistically relevant differences among biomarkers levels should be discussed more in depth; are the differences in IL1, IL6 and TNFa levels also clinically/biochemically relevant? What seems to be proven are the antioxidants/antinflammatory effects of Mangiferin, without a clear effect on myocardial function recovery/myocarditis healing

MINOR ISSUES

-   The clinical relevance of the difference in thickness of LV posterior wall seems minimal; tis should be highlighted when discussing this finding, which seems of minor relevance.

-   In the study conclusion, the Authors state that this study supports the relevance of echo in evaluating murine models of myocarditis. Actually, this was out of the scope of the study, also considering that echo was not compared to other modalities of evaluation of LV function and its role has already been investigated in larger studies. Please erase this section both in the abstract and conclusions

-   Line 89. “Autoimmune myocarditis also known as giant cell myocarditis” (GCM). This is a mistake: GCM is only one of the possible histological types of autoimmune myocarditis, and one of the most rare ones; lymphocytes autoimmune myocarditis is far more common

-   Fig 1. Please specify that this is only one of the possible cascades leading to fibrosis.

-   The methods section should come before the results

-   Were echos always performed by the same operator, or by different ones?

-   How was TAC exactly assessed?

-   Tab 1. 1) Please show the p-values in the tab, for clarity. 2) The variables are reported as mean and SD; were parametric test used for comparison?

-   In the discussion section, in addition to oxidative stress and RS; please briefly discuss the role of innate immunity in myocarditis (see: Vicenzetto C et al. The Role of the Immune System in Pathobiology and Therapy of Myocarditis: A Review. Biomedicines. 2024)

-   Line 336: “increased LVEF” respect to what? Please specify this and insert a brief summary of the results

-   Line 345. Why did the Authors decide to use a different dosage of the drug compared to previous literature?

Author Response

Dear Reviewer,

The authors of the original article “EFFECTS OF MANGIFERIN IN EXPERIMENTALLY INDUCED MYOCARDITIS IN RATS” express deep gratitude to you for agreeing to review our work. Thank you very much for the critical, but professional analysis of our manuscript and for the remarks made, which we gratefully accept. The authors have made the necessary changes to the text in accordance with the comments made. The changes made have been marked in red in the new version of the manuscript.

Reviewer 2

MAJOR ISSUES

Comment 1:   The fact that LVEF was not significantly lower in the myocarditis untreated group than the other groups (only a “trend” was observed, line 176) should be discussed more in depth. This is likely the main reason why echocardiographic features were not significantly improved by the different treatments (again, only a “trend” towards LVEF improvement was observed, line 177). To be sure that myocarditis was correctly induced in all groups, did some animal undergo myocardial biopsy before the end of the experiment (i.e. day 2)? In addition: “Line 280. In 1 case no inflammatory infiltrates were observed.”  This raises serious concerns about the fact that myocarditis was really induced in these models. Spontaneous healing of myocarditis is a likely reason, but it is not discussed here; actually, this may happen in >50% of human myocarditis, without any treatment. In my opinion, the Authors should focus on this aspect to reinforce the value of the overall findings.

Response 1: Thank you for the strong and well-reasoned comment. We have attempted to revise the text to address the points mentioned below.

We considered the trend of the obtained values, although they did not have statistical significance, with the main limitation being the sample size. Additionally, the ejection fraction is an observer-dependent measurement and difficult to quantify in rats due to their small heart size and increased heart rate. Thus, for our study, we considered echocardiography to see if there are indeed changes that could justify its use in the evaluation of experimentaly induced myocarditis. Otherwise, echocardiography is not as accurate throughout measuring LVEF. The most accurate investigation  is cardiac MRI—a method to which we did not have access during this study. Accordingly we have put in the text : We considered as trend the obtained values, although they did not have statistical significance, with the main limitation being the sample size. Additionally, the LVEF is an observer-dependent measurement and difficult to quantify in rats due to their small heart size and increased heart rate, so obtaining an optimal image and calculating diameters, volumes, with the axes as equal as possible, are considered challenging aspects.  Echocardiography was merely a suggestive tool for assessing induced acute myocarditis, not a diagnostic one in our study, and is not as accurate for quantifying kinetic impairment resulting from inflammation, edema or fibrosis, aspects that could be more accurately described by cardiac MRI.

Biopsies were not performed on day 2 after myocarditis induction due to the small number of animals used, which did not allow for this. However, the presence of changes in 4 out of 5 animals (one of the animals in the positive control group was found dead in the first week of the experiment and the quality of the tissue sample has been affected) in which myocarditis was induced and no treatment administered,  was considered representative for the positive control group. This was a risk  assumed until the end of the experiment, which we have to consider an obvious limitation.

Comment 2: There is a lack of consideration of recent literature clearly supporting the use of steroids in human virus-negative myocarditis; a large amount of literature has been published showing that steroid therapy has short and long-term benefits in terms of both overall survival and strong clinical outcomes such as LVEF improvement (see: 1) Chimenti Cet al. Immunosuppressive therapy in virus-negative inflammatory cardiomyopathy: 20-year follow-up of the TIMIC trial. Eur Heart J. 2022, 2) Caforio ALP et al. Long-term efficacy and safety of tailored immunosuppressive therapy in immune-mediated biopsy-proven myocarditis: A propensity-weighted study. Eur J Heart Fail. 2024); steroid therapy has also been included among the latest ESC recommendations in heart failure in humans. Please amend erroneous sentences both in the abstract (lines 36-37), introduction (line 126) and discussion sections.

Response 2: We appologies for the missunderstanding. We have revised the text highlighting that our results demonstrate their beneficial effects—evidenced by the absence of histopathological changes, at the end, in the treated groups. However, we aimed to emphasize that the mortality in that group could have been influenced by other adverse effects associated with Prednisone, unrelated to the evolution process of myocarditis. We have updated the text accordingly. In the abstract:

Corticosteroids are frequently employed in the treatment of autoimmune myocarditis and appear to lower mortality rates compared to conventional therapies for heart failure. 

In the introduction section:

Some studies showed that administration of corticosteroids has not been associated with a significant reduction in mortality compared to conventional heart failure treatment, although may improve left ventricular (LV) function. However, current studies have shown that standard immunosuppressive therapy is recommended to be administered long-term and tailored to the patient. Caforio and collaborators, in their recent study results, support the effectiveness and safety of such therapy in immune-mediated myocarditis.

In discution section :

The cause of death remains unclear, although myocarditis was excluded as the determining factor based on histopathological results. It is important to note that this could be a limitation of glucocorticoid administration in experimental models, potentially due to their other effects rather than their immunosuppressive action.

This could especially impact the reliability of conclusions drawn from subgroups, such as those treated with Prednisone, where significant mortality occurred without a clear explanation. This highlights the importance of carefully considering potential side effects when evaluating treatment outcomes. In our study, mortality could introduce bias and influence the interpretation of results, particularly in assessing the efficacy and safety of Prednisone.

Comment 3: The statistically relevant differences among biomarkers levels should be discussed more in depth; are the differences in IL1, IL6 and TNFa levels also clinically/biochemically relevant? What seems to be proven are the antioxidants/antinflammatory effects of Mangiferin, without a clear effect on myocardial function recovery/myocarditis healing.

We thank the reviewer for the observation. The observed variations in IL-1, IL-6, and TNF-α levels across our experimental groups reflect the degree of inflammation and immune response, potentially indicating either the intensity or resolution of these processes. While our statistically significant results suggest that these differences are unlikely to be attributable to chance, we were unable to directly establish their clinical or biochemical significance based on limited number of animals in our study. It is essential to evaluate whether these biomarker fluctuations have substantive implications for patient outcomes or disease progression. To confirm their utility as prognostic indicators or to assess treatment efficacy, long-term studies are required to validate their potential in clinical settings.

MINOR ISSUES

Comment 4:   The clinical relevance of the difference in thickness of LV posterior wall seems minimal; tis should be highlighted when discussing this finding, which seems of minor relevance.

Response 4: Thank you for your sugestion. We have enhanced the text as follows: Left ventricular wall thickening in the setting of acute myocarditis is usually transient and was reported in few studies during the last decades. These studies demonstrated that the thickenss of the posterior wall is greatest in the day 1-3 after myocarditis onset and improves to near normal during convalescent phase. Other studied supported the hypothesis that this process is caused by interstitial edema. As we have demonstrated that Trolox has a lower effect comparing to Mangiferin on reducing the inflammatory process, the persistence of increased thickness of the posterior LV wall on day 21 could be attributed to persistent inflammation.

Comment 5:  In the study conclusion, the Authors state that this study supports the relevance of echo in evaluating murine models of myocarditis. Actually, this was out of the scope of the study, also considering that echo was not compared to other modalities of evaluation of LV function and its role has already been investigated in larger studies. Please erase this section both in the abstract and conclusions.

Response 5: We thank the reviewer for the suggestion. We have erased acccordingly the mentioned section from the abstract : The value of echocardiography for assessing myocarditis in experimental model, was highlighted, despite some limitations. And we have reformulated the final conclusion.

Comment 6:   Line 89. “Autoimmune myocarditis also known as giant cell myocarditis” (GCM). This is a mistake: GCM is only one of the possible histological types of autoimmune myocarditis, and one of the most rare ones; lymphocytes autoimmune myocarditis is far more common

Response 6: Thank you for pointing this out and apologies for this oversight. We have, accordingly, erase from the text:  also known as giant cell myocarditis.

Comment 7:  Fig 1. Please specify that this is only one of the possible cascades leading to fibrosis.

REsponse 7: Thank you for pointing this out. We have changed the short phrase in the text with:  One of the multiple pathways of the fibrosis process is depicted schematically in Figure 1. ; also, in the legend we put: Figure 1. A proposed schematic representation ...

Comment 8: The methods section should come before the results.

Response 8: Apologies for this oversight, we have swich the order in the text, as you suggested.

Comment 9: Were echos always performed by the same operator, or by different ones?

Response 9: Echocardiography was performed by two operators, with both present at each examination. we have put in the text also, in section 2.3.

Comment 10:  How was TAC exactly assessed?

Response 10: We thank you for the observation. The total antioxidant capacity (TAC) was determined as a global index of antioxidant defense. It was measured using a colorimetric method based on the neutralization of dianisidyl radicals resulting from the oxidative process of ortho-dianisidyl. Therefore, a standard solution of Fe2+-o-dianisidyl underwent the Fenton reaction with a standard solution of H2O2, forming hydroxyl -OH radicals. These radicals, in the presence of an acid, oxidized o-dianisidines to dianisidyl radicals. The antioxidant agents in the sample inhibited the oxidation reactions and the appearance of coloration. At the end of the reactions, the color intensity was determined spectrophotometrically. This assay was calibrated using TX and results were expressed as mM TE/L.

Comment 11: Tab 1. 1) Please show the p-values in the tab, for clarity.

Response 11:

1) Thank you for the suggestion. The p values for each comparison are labeled with a symbol. Because we did multiple comparisons in the same table, p values written in the tab for each comparison would create difficulties in reading the results. We did a legend under each table where we explained the significance of each symbol and the p-values corresponding to them.

Values are expressed as mean ± SD (standard deviation). * vs. CONTROL: * p<0.05, ** p<0.01, *** p<0.001; # vs. MYO: # p<0.05, ## p<0.01, ### p<0.001; f vs. MYO+Prednisone: f p<0.05, ff p<0.01, fff p<0.001; a vs. MYO+Trolox: a p<0.05, aa p<0.01, aaa p<0.001; MYO – Myosin; TAC -Total antioxidant capacity; NOx – Nitrites and nitrates;

2) The variables are reported as mean and SD; were parametric test used for comparison?

2)We thank you for the observation. Yes, parametric tests were used because data distribution was normal. In order to check the normality of the data distribution we have used the Kolmogorov-Smirnov and Shapiro-Wilk tests. We added them in the statistical methods.

Comment 12:  In the discussion section, in addition to oxidative stress and RS; please briefly discuss the role of innate immunity in myocarditis (see: Vicenzetto C et al. The Role of the Immune System in Pathobiology and Therapy of Myocarditis: A Review. Biomedicines. 2024)

Response 12: Thank you for the sugestion. We have added in introduction section the following text, based of the recomended article: Myocarditis involves both innate and adaptive immune responses, with each contributing to different phases of the disease. The acute phase is driven by the innate immune system, particularly through cytokine and chemokine release, while the subacute and chronic phases engage the adaptive immune system, including T cells. If inflammation is not fully resolved, the disease may progress to a chronic condition, leading to dilated cardiomyopathy (DCM) and heart failure. Various immune cells, such as macrophages and T-helper cells, play key roles in this progression, with their effects modulated by cytokines and other signaling molecules. Despite advances, the complete immunological mechanisms of myocarditis are not fully understood, warranting further research to develop targeted therapies.

Comment 13:   Line 336: “increased LVEF” respect to what? Please specify this and insert a brief summary of the results

Response 13: Thank you for the comment. We have reformulated the text as follows: These changes could be likely attributable to the reduction in inflammatory cell infiltration, resulting in the normalization of ventricular contraction. This is evidenced by increased LVEF and LVFS, along with decreased LVD in the treated groups, particularly evident on the 21 day of evaluation. Such findings may indicate a positive outcome in preventing left ventricular remodeling and the progression to dilated cardiomyopathy and heart failure

Comment 14:  Line 345. Why did the Authors decide to use a different dosage of the drug compared to previous literature?

Response 14: When we organized the study protocol, we aimed to find studies that mentioned a dosage capable of demonstrating both antioxidant and anti-inflammatory effects, with suggested minimum doses starting at 10 mg/kg. (Lum PT et al.2023). As this is, to our knowledge, the first study evaluating the effects of Mangiferin in experimentally induced autoimmune myocarditis, we sought to achieve both effects. In the study cited in the discussion chapter, the dosage was indeed 40 mg/kg, but the administration duration was 15 days, which is shorter than our study's 21-day duration, for that reason we have reformulated the text as following:  For the study protocol, we aimed to select a dose of Mangiferin that would achieve both its antioxidant and anti-inflammatory effects. According to the literature, the minimum dose reported to produce these effects is 10 mg/kg ( Lum et colab). On the other hand, the milder effects of Mangiferin on systolic function parameters in our study may be attributed to the lower dose used (25 mg/kg), compared to higher doses (40 mg/kg) that have significantly improved these parameters in rats with myocardial ischemia-reperfusion (IR) injury. This comparison does not take into account the cumulative effects of Mangiferin, which was administered for 21 days in our study, versus the 15-day administration of higher doses in the previously mentioned study.

We appreciate your input, which has helped us improve the clarity and completeness of our work.

Round 2

Reviewer 2 Report

Comments and Suggestions for Authors

The Authors responded to my comments in a satisfactory manner.

Author Response

Dear Reviewer, thank you for the revised manuscript. All the specific suggestions you previously addressed have been incorporated into the text. Thank you once again, and please let us know if any further changes are needed.